

# On the non-linear response of Antarctic ice shelf surface melt to warming

Marte Gé Hofsteenge[1], Willem Jan van de Berg[1], Christiaan van Dalum[1,2], Kristiina Verro[1,3], Maurice van Tiggelen[1], and Michiel van den Broeke[1]

[1]Institute for Marine and Atmospheric Research (IMAU), Utrecht University, Utrecht, the Netherlands
[2]Royal Netherlands Meteorological Institute, De Bilt, the Netherlands
[3]National Centre for Climate Research, Danish Meteorological Institute, Copenhagen, Denmark

**Correspondence:** Marte Gé Hofsteenge (m.g.hofsteenge@uu.nl)

**Abstract.** Surface meltwater can saturate firn, form melt ponds, and trigger hydrofracturing of Antarctic ice shelves, ultimately accelerating grounded ice flow and contributing to sea level rise. Although the response of surface melt to atmospheric warming (expressed by near-surface air temperature) is known to be non-linear, the mechanisms driving this non-linearity remain poorly understood. In this study we explain the non-linear temperature-melt relationship from an energy balance perspective and assess its spatial variability across Antarctic ice shelves. We use the regional climate model RACMO2.4p1, forced by ERA5 re-analysis and two global earth system models under the SSP3-7.0 high emission scenario, to simulate contemporary and future Antarctic climate and surface mass balance until 2100. We find that the temperature dependence of net shortwave radiation is the primary driver of the non-linearity. On relatively cold ice shelves, warming increases cloud cover and snowfall, raising albedo, reducing net shortwave radiation. In contrast, on warmer ice shelves the snowmelt-albedo feedback dominates the response: warming leads to melt that reduces albedo, enhancing shortwave radiation absorption. The temperature–melt relationship also varies spatially: ice shelves in drier regions experience more melt at the same average summer temperatures than those in wetter regions, highlighting the role of snowfall in suppressing the albedo feedback. When mean summer air temperatures reach or exceed the melting point (0 °C), ice shelves becomes even more sensitive to warming. Surface temperatures can not rise above 0°C while the atmosphere can, allowing the sensible heat and net longwave radiation to increase. At the same time, snowfall transitions to rain, amplifying the albedo feedback. Our results suggest that currently colder, drier and stable ice shelves could experience rapid increases in melt under future warming, with implications for their long-term stability.





## 1 Introduction

Surface melt plays a critical role in the stability of Antarctic ice shelves, which act as barriers slowing the flow of inland ice
into the ocean (Fürst et al., 2016). Episodes of intense or prolonged surface melting can lead to the formation of melt ponds
on the surface (Kingslake et al., 2017), which may trigger hydrofracture and rapid ice shelf collapse (Kuipers Munneke et al.,
2014; Lai et al., 2020). Such collapses reduce ice shelf buttressing, accelerate flow and mass loss of grounded ice and result
in sea level rise (Bell et al., 2018). Surface melt and runoff from Antarctic ice shelves is projected to increase with warming
(Gilbert and Kittel, 2021; Kittel et al., 2021; Ligtenberg et al., 2013). However, the relationship between temperature and melt
is highly non-linear (Abram et al., 2013), resulting in a large spread of predicted melt rates at the end of the 21st century
between different future scenarios (Trusel et al., 2015; Seroussi et al., 2020).

Polar-adapted regional climate models (RCMs), such as the polar (p) version of the Regional Atmospheric Climate Model
(RACMO), are currently among the most effective tools for quantifying current and predicting future Antarctic surface melt.
These models include snowpack physics and surface energy balance (SEB) schemes to simulate key surface processes driving
melt, and are typically run at much higher spatial resolutions (1-20 km) than global Earth system models ($\sim 100$ km). However,
due to the computational costs, RCMs are not typically used to estimate melt in large ensembles of future climate scenarios,
such as those from CMIP, or coupled to ice sheet models to link the surface processes and ice dynamics. Instead, studies pro-
pose simplified empirical relationships or melt potential indices based solely on near-surface air temperature (Orr et al., 2023;
Trusel et al., 2015; Vaughan, 2006; Zheng et al., 2023). These approaches rely on the fact that many SEB components are them-
selves dependent on temperature, such as incoming longwave radiation and sensible heat flux (Ambach, 1988; Braithwaite and
Olesen, 1990; Ohmura, 2001). While widely used (Coulon et al., 2024; DeConto et al., 2021; Golledge et al., 2019), empirical
relationships between temperature and melt such as positive degree day (PDD) models or temperature-melt index models, are
often applied uniformly across space. This is despite the known spatial variability in melt sensitivity to temperature, which
arises from differences in local surface conditions, cloudiness, and SEB regimes across Antarctic ice shelves (van den Broeke
et al., 2023; Zheng et al., 2023).

The strong relationship between summer near-surface air temperature and surface melt on Antarctic ice shelves can be
characterized by an exponential relationship (Trusel et al., 2015; van Wessem et al., 2023). This means that for ice shelves
already experiencing melt, even a small temperature increase can lead to a disproportionately large increase in melt, potentially
reaching levels of melt at which ice shelves have collapsed in the past (Scambos et al., 2000; Trusel et al., 2015; van den Broeke,
2005). The strong non-linearity in this relationship has been linked to the snowmelt–albedo feedback (Jakobs et al., 2019), but
the extent to which this mechanism, or other temperature-sensitive SEB components contribute to the observed non-linearity
remains unclear. In this study we use output from RACMO to investigate the physical processes driving the non-linearity of
the relationship between summer air temperature and surface melt across Antarctic ice shelves. We analyse both historical
and future simulations to 1) assess the spatial variability in the temperature-melt relationship and 2) identify the dominant



SEB components contributing to the non-linearity. Our findings will provide new insights into the physical controls on melt sensitivity and provide guidance on when and where temperature-based approaches to estimate surface melt are appropriate, and where they lack reliability due to more complex local SEB conditions.

## 2 Methods

### 2.1 The regional climate model RACMO

The Regional Atmospheric Climate Model (RACMO) is a hydrostatic regional climate model developed at the Royal Netherlands Meteorological Institute (KNMI). It incorporates atmospheric dynamics from HIRLAM (Undén et al., 2002), which uses a semi-Lagrangian approach with semi-implicit time stepping, and physical parameterizations from the ECMWF Integrated Forecasting System (IFS) (ECMWF, 2020). We use the polar version of RACMO which is developed and maintained at the Institute for Marine and Atmospheric Research Utrecht (IMAU). This version is specifically adapted for simulating the climate and surface processes of the polar regions and includes a multi-layer snow/firn model applied to glaciated surface tiles. The snow model simulates key processes to represent the surface energy and mass balance of snow, such as metamorphism, compaction, melt and refreezing of snow. In this study, we used the latest RACMO version (2.4p1), in this paper referred to as RACMO. The main differences between RACMO2.4p1 and the previous operational version, RACMO2.3p2, are (van Dalum et al., 2024):

1. Update of IFS physics parameterizations from ECMWF cycle 33r1 to 47r1 (ECMWF, 2020), which includes improvements of cloud and precipitation physics such as a better representation of mixed phase clouds, separate prognostics variables for cloud water/ice, rain and snow that allows for advection of precipitation, the radiation scheme is replaced by ecRad, aerosols prescription with Copernicus Atmospheric Monitoring Service (CAMS), and improved cloud optical properties from the Suite Of Community RAdiative Transfer codes based on Edwards and Slingo (SOCRATES). Additional processes that are included in the change to IFS cycle 47r1 are described in van Dalum et al. (2024);

2. Fractional ice cover, based on the BedMachine Antarctica version 3 ice mask (Morlighem et al., 2020), where part of grid cells can be partially glaciated and therefore better representing areas such as the McMurdo Dry Valleys ;

3. A narrowband snow albedo model, which was introduced in the non-operational RACMO version 2.3p3 (van Dalum et al., 2022). This new albedo model explicitly resolves radiation penetration and subsurface heating in snow and ice;

4. An updated snowdrift model (Gadde and van de Berg, 2024)





## 2.2 Surface energy balance

Within the updated snow model, shortwave radiation can penetrate in the snowpack and be absorbed in the subsurface. The total
net absorbed shortwave radiation ($SW_{net}$) is thus split up in surface absorption ($SW_{net, surf}$) and internal absorption ($SW_{pen}$).
Internal shortwave absorption leads to warming of the snowpack and internal melt when the snow reaches the melting point.
As a result, melt can occur both at the surface and in the snowpack. The SEB is defined as:

$$Q_{M, surf} = LW_{net} + SW_{net, surf} + SH + LH + Q_G \tag{1}$$

where $Q_{M, surf}$ is the energy used for surface melt, $LW_{net}$ is net longwave radiation, SH and LH represent the turbulent fluxes
of sensible and latent heat and $Q_G$ is the subsurface conductive heat flux. All fluxes are defined as positive towards the surface.

Since we are interested in understanding the relationship between temperature and total melt; i.e. occurring both at the
surface as well as in the subsurface, we use a pseudo-SEB for the analysis which is not strictly valid for a skin layer as in
Eq. (1), but for the near-surface. This definition is comparable to previous SEB formulations, in which penetration of solar
radiation was not included, and is as follows:

$$Q_M = LW_{net} + SW_{net} + SH + LH + Q_{G,rec}, \tag{2}$$

Where $Q_M$ is the energy used for melt (surface + internal melt) and $Q_{G, rec}$ is a reconstructed conductive heat flux defined
as $Q_{G, rec} = Q_G + Q_{m, internal} - SW_{pen}$, with $Q_{m, internal}$ energy used for internal melt. This formulation assumes that, on seasonal
timescales, the effects of short-term heat storage and energy redistribution in the near-surface are negligible, such that total
SWnet (surface + penetrated) can be used as the radiative driver of total melt. In Appendix A we show that this pseudo-SEB
produces comparable SEB terms to the classical formulation without shortwave penetration, supporting its use for the analysis
in this study.

The surface mass balance (SMB) is defined as the difference between accumulation and ablation in the near-surface firn or
ice:

$$SMB = P_{tot} - SU_s - SU_{ds} - RU - ER \tag{3}$$

with $P_{tot}$ total precipitation, $SU_s$ is surface sublimation and $SU_{ds}$ sublimation of blowing snow, RU runoff and ER drifting
snow erosion which can be both ablative (erosion) and accumulative (deposition). RU is simulated when the snow layers
become saturated with meltwater or rain.



## 2.3 Historical and future forcing

We use RACMO2.4p1 with a domain covering Antarctica and the southern tip of South America with a horizontal resolution of 11 km, forced with both reanalysis and Earth-System-Model (ESM) datasets. The simulations for this domain have been done as part of the PolarRES project, an EU Horizon 2020 funded project that uses RCMs to simulate the current and future climate of the polar regions. We use the RACMO simulation forced by ERA5 reanalysis data (1979-2023), previously evaluated with observations by van Dalum et al. (2025), as reference simulation for the historical period, which is hereafter referred to as RACMO(ERA5). Future projection simulations (2015–2099) were performed using boundary conditions from two CMIP6 ESMs: the Community Earth System Model 2 (CESM2) and the Max-Planck Institute Earth System model (MPI-ESM), under the high emission SSP3-7.0 scenario.

As a second evaluation step, historical simulations (1985-2014) with these ESMs are compared to RACMO(ERA5). The RACMO simulations with ESM forcing are hereafter referred to as RACMO(CESM2) and RACMO(MPI-ESM). CESM2 and MPI-ESM were selected within the PolarRES project from the CMIP6 ESMs using a storyline approach to represent two contrasting but plausible Antarctic climate futures; CESM2 reflects a future with extensive sea ice loss and an earlier summertime stratospheric polar vortex breakdown, while MPI-ESM captures a scenario with limited sea ice loss and a delayed polar vortex breakdown (Williams et al., 2024).

At the lateral boundaries of the modelling domain, windspeed, temperature, pressure and humidity are specified using multi-level data from either ERA5 or ESM output. Sea surface temperature and sea-ice cover are defined at the ocean surface boundary and prescribed from ERA5 or ESM output. Sea ice temperature is calculated by a four-layer sea ice slab model with a fixed thickness of 1.5 m. Furthermore, in the free atmosphere the temperature, wind field and humidity are gently relaxed to either the ERA5 or ESM output, following van de Berg and Medley (2016)

## 3 Results

### 3.1 Evaluation of RACMO historical simulations with ESM forcings

Before using RACMO for future scenario simulations, we need confidence in the model performance when forced by ESMs. To assess this, we compare the average climate in the historical simulation (1985-2014) with that in RACMO(ERA5), since direct comparison with in situ observations is not possible for ESM output unconstrained by data assimilation. RACMO(ERA5) has been extensively evaluated against weather station and mass balance observations (van Dalum et al., 2025). The updated version 2.4p1 performs well in simulating Antarctica's surface mass balance, temperature, albedo, and shortwave radiation, whereas longwave radiation and turbulent fluxes still require improvement. Figure 1 shows the differences in annual means of several key variables between simulations forced by ESMs and forced by ERA5. RACMO(MPI-ESM) is warmer compared to RACMO(ERA5), especially over the sea ice zone and in the Ross Sea and Wedell Sea sectors (Fig. 1a). This coincides with





lower sea ice extent in RACMO(MPI-ESM) compared to RACMO(ERA5) (Fig. 1b). While the warm bias is smaller over the Antarctic continent, RACMO(MPI-ESM) is still significantly warmer than RACMO(ERA5) in Dronning Maud Land, the coast of Marie Byrd Land and other coastal regions in East-Antarctica. The temperatures and sea ice conditions in RACMO(CESM2) are more realistic, but the simulations have a cold bias over West Antarctica and large parts of East-Antarctica, where the 500 hPa geopotential height is lower compared to RACMO(ERA5).

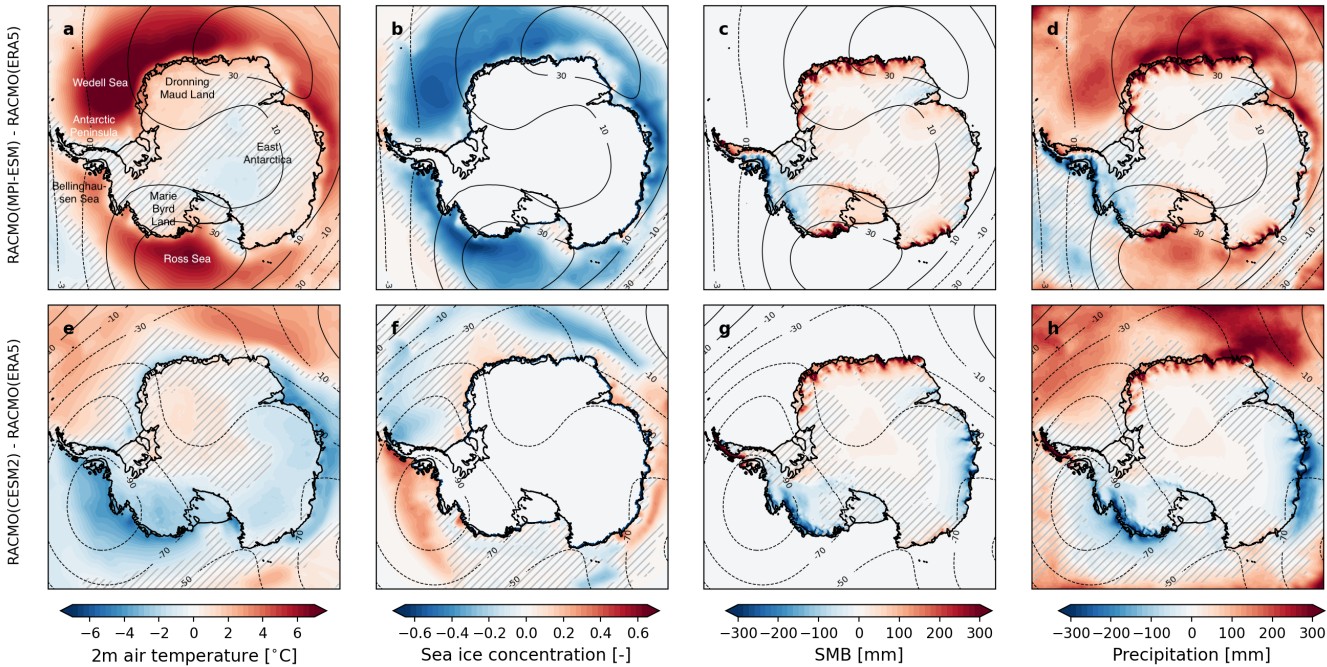

**Figure 1.** Difference in annual mean near-surface air temperature, sea ice concentration, surface mass balance, and precipitation between RACMO simulations forced by MPI (a-d) or CESM (e-h) and those forced by ERA5 in the period 1985-2014. Hatched areas show where differences are smaller than the standard deviation over the historical period in ERA5. Contourlines indicate corresponding difference in 500hPa geopotential height with RACMO(ERA5).

RACMO(CESM2) and RACMO(MPI-ESM) have similar SMB anomaly patterns, with more positive SMB in Dronning Maud Land and a more negative SMB in West Antarctica compared to RACMO(ERA5) (Fig. 1c,g), driven by differences in precipitation (Fig. 1d,h). The negative SMB anomaly in East-Antarctica in RACMO(CESM2), which is due to an underestimation of precipitation, is not present in RACMO(MPI-ESM). Over the Antarctic Peninsula, RACMO(MPI-ESM) simulates less precipitation on the western side and more on the eastern side compared to RACMO(ERA5). This pattern is associated with a weaker Amundsen Sea Low, reflected in a positive geopotential height anomaly over the Amundsen-Ross Sea in Fig. 1d. In contrast, RACMO(CESM2) shows lower geopotential heights over the Bellinghausen Sea relative to RACMO(ERA5), which



enhances westerly winds across the Antarctic Peninsula and orographic precipitation on the western slopes.

In this study we focus on melt, which occurs almost exclusively during the summer (DJF) season, so from here onwards we focus on the simulations in summer. Table 1 compares average summer near-surface air temperature, energy and mass balance terms over the Antarctic ice sheet. The simulations show good agreement in summer air temperature and most SEB terms. The

underestimation of sea ice and warm bias over the sea ice zone and coastal regions is much smaller in summer (Appendix Fig. B1) compared to the annual mean that was shown in Figure 1.

**Table 1.** RACMO2.4p1 simulated summer averages (1985-2014) forced by ERA5, CESM2 and MPI-ESM. The table shows 30-year mean values and standard deviations for ice sheet average near-surface air temperature, surface energy balance [W m$^{-2}$] and ice sheet integrated average summer totals of mass balance components [Gt summer$^{-1}$].

|  | **RACMO(ERA5)** | **RACMO(CESM2)** | **RACMO(MPI-ESM)** |
|---|---|---|---|
| $T_{2m}$ [°C] | -23.2 ± 0.8 | -24.0 ± 0.6 | -23.8 ± 0.6 |
| $SW_{in}$ [W m$^{-2}$] | 349.7 ± 1.9 | 350.6 ± 1.4 | 348.7 ± 2.0 |
| $LW_{in}$ [W m$^{-2}$] | 155.0 ± 2.5 | 151.3 ± 1.9 | 154.3 ± 2.0 |
| $SW_{net}$ [W m$^{-2}$] | 58.4 ± 0.9 | 57.9 ± 1.3 | 56.6 ± 1.0 |
| $LW_{net}$ [W m$^{-2}$] | -63.7 ± 0.9 | -64.0 ± 0.8 | -62.3 ± 0.9 |
| SH [W m$^{-2}$] | 9.6 ± 0.5 | 10.2 ± 0.6 | 9.9 ± 0.3 |
| LH [W m$^{-2}$] | -1.9 ± 0.1 | -1.8 ± 0.1 | -1.9 ± 0.2 |
| Precipitation [Gt summer$^{-1}$] | 565 ± 63 | 556 ± 68 | 608 ± 45 |
| Surface sublimation [Gt summer$^{-1}$] | 70.8 ± 5.5 | 67.8 ± 5.6 | 72.1 ± 6.1 |
| Snowdrift sublimation [Gt summer$^{-1}$] | 24.8 ± 3.8 | 28.3 ± 4.2 | 27.6 ± 2.5 |
| Melt [Gt summer$^{-1}$] | 115 ± 38 | 71 ± 25 | 120 ± 41 |
| Refreezing [Gt summer$^{-1}$] | 110 ± 37 | 67 ± 25 | 116 ± 39 |
| Runoff [Gt summer$^{-1}$] | 4.3 ± 3.5 | 4.7 ± 4.0 | 5.0 ± 3.1 |
| SMB [Gt summer$^{-1}$] | 461 ± 62 | 455 ± 66 | 503 ± 47 |

Overall, the dynamically downscaled SEB and SMB in summer from RACMO(CESM2) and RACMO(MPI-ESM) match well with that of RACMO(ERA5). The largest differences in SMB terms between the simulations lie in melt and precipitation.

Melt is comparable between RACMO(ERA5) and RACMO(MPI-ESM), but is underestimated in RACMO(CESM2), likely due to a cold bias especially in West-Antarctica. Precipitation in RACMO(MPI-ESM) is higher than in RACMO(ERA5) but the difference is still smaller than typical variability in summer precipitation. The difference in precipitation is dominated by positive anomalies on the western side of the Antarctic Peninsula and over the coastal region of Dronning Maud Land (Appendix Fig. B1).




### 3.2 Historical and future trends in Antarctic warming and surface melt

In our simulations under the SSP3-7.0 scenario, warming of the near-surface atmosphere over the 21st century is strongest in autumn and winter, especially over the sea ice zone (Fig. 2). We find large differences between RACMO(CESM2) and RACMO(MPI-ESM): the former consistently shows stronger warming across all seasons, with the most pronounced differ-
ences in autumn and winter that have substantial sea ice decline. Part of the different warming rates stem from the different starting conditions in the historical climate states, with MPI-ESM simulating significantly lower sea ice concentrations and higher air temperatures (Fig. 1b). Another reason for the difference in warming is likely the contrasting atmospheric circulation response of the ESMs. MPI-ESM has a stronger intensification and poleward shift of the jet stream compared to CESM2, which limits the advection of warm air from lower latitudes and therefore suppresses warming, especially in winter (Williams
et al., 2024).

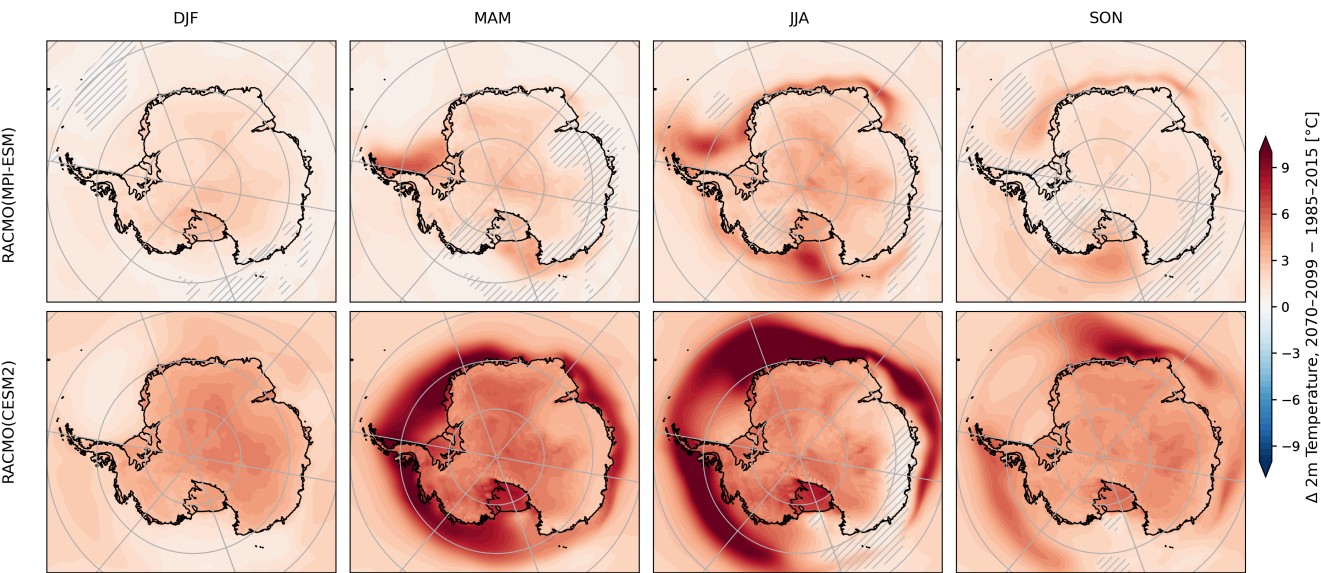

**Figure 2.** Average near-surface air temperature change between the current climate (1985–2014) and future climate (2070–2099) as modeled by RACMO. Panels (a–d) are forced by MPI-ESM, while panels (e–h) are forced by CESM2. Hatched areas show where changes in temperature are smaller than the standard deviation over the historical period in ERA5.

The warming over the Southern Ocean is weakest in summer, when sea ice concentration, and thus trends in sea ice concentration, are smallest. The large thermal inertia of the ocean inhibits a fast surface warming and subsequently a quick rise of the near-surface air temperature. Nevertheless, both RACMO(CESM2) and RACMO(MPI-ESM) show warming over the entire
Antarctic continent during summer. When focusing on near-surface air temperature over ice shelves, we find good agreement between the historical simulations of near-surface temperature, both in terms of the mean and inter-annual variability (Fig. 3).



The largest differences are found on the ice shelves of West Antarctica (Fig. 3g), with RACMO(CESM2) being colder and RACMO(MPI-ESM) warmer compared to RACMO(ERA5). For all regions, ice shelves are warming significantly over the 21st century and consistently stronger in RACMO(CESM2) compared to RACMO(MPI-ESM). Interestingly, for most regions

melt in RACMO(MPI-ESM) and RACMO(CESM2) is more similar in the historical period and diverges in the future scenario, apart from ice shelves in West Antarctica (Fig. 3g) that start with a large difference but converge in the future.



**Figure 3.** Timeseries of summer average near-surface air temperature and melt over all ice shelves and subgroups of ice shelves, where AP represents the Antarctic Peninsula, EAIS is East Antarctica, WAIS is West Antarctica, and COLD includes the Filchner-Ronne and Ross Ice Shelves. Error bars indicate the mean and standard deviation over the historical period (1985–2014). Trendlines are shown where a significant trend (p-value < 0.05) is detected in either the historical (1985–2014) or SSP3-7.0 (2015–2099) simulations.

No significant trends in melt are found over the historical period, except for a decrease in melt on the ice shelves of the Antarctic Peninsula in RACMO(ERA5) (Fig. 3d). This agrees with previous studies that report a significant regional cooling





since the late 1990s driven by changes in atmospheric circulation and increased sea ice advection (Turner et al., 2016; van Wessem et al., 2016), which changed in the mid-2010s together with changes in the large-scale climate modes (Carrasco et al., 2021). Melt starts to increase in the future simulations, showing significant increases over all regions. On the Filchner-Ronne and Ross Ice Shelves, the interannual variability relative to the overall melt trend is largest, particularly after 2070. Melt increases more rapidly in RACMO(CESM2) compared to RACMO(MPI-ESM) for all regions, consistent with the stronger

temperature increase. As a result, for most ice shelves RACMO(CESM2) starts with lower temperatures and melt, but ends higher than RACMO(MPI-ESM)

### 3.3 Spatial variability in temperature-melt relationship

Given the strong link between temperature and melt suggested by previous research (e.g. van Wessem et al. (2023); Trusel

et al. (2015)) and the future trends (Fig. 3), we now explore the temperature–melt relationship in more detail, showing that it is highly non-linear and varies significantly across regions (Fig. 4). Even though the simulations have different warming rates, the relationship between temperature and melt is consistent across the ERA5 and ESM model forcings (scatter plots). The Filchner-Ronne and Ross ice shelves are coldest and while they experience significant warming, their melt rates remain small compared to other ice shelves, with shelves on the Antarctic Peninsula experiencing most melt. Summer melt is exponentially

related to summer air temperature (Kittel et al., 2021; van Wessem et al., 2023), although the shape and steepness of this relationship vary across different ice shelves (Fig. 4).

Ice shelves in drier climates tend to experience more melt at the same temperatures compared those in wetter climates. The boxplots in Fig. 4 illustrate the relative dryness of the highlighted ice shelves. For example, based on the exponential fits, a

summer average air temperature of -4°C would lead to around 390 mm melt at the Amery Ice Shelf, that receives little snowfall, compared to 90 mm at Nickerson Ice Shelf where snowfall rates are higher. This shows that a given melt rate is reached at different air temperatures across the ice shelves. For instance, reaching 200 mm melt occurs between -6 °C and -2.5 °C, depending on the ice shelf. This temperature is lower for ice shelves with in drier climates (Appendix Fig. C1).

### 3.4 Variable albedo feedbacks

To better understand the relationship between temperature, surface melt, and the role of snowfall, we examine one of the primary drivers of Antarctic surface melt: the absorption of shortwave radiation (Elvidge et al., 2020; Gilbert et al., 2022; Hofsteenge et al., 2023; Jakobs et al., 2020). Figure 5 illustrates the temperature dependency of summer albedo across the major Antarctic ice shelves (ice shelf area > 800 km$^2$). On cold ice shelves (where summer air temperatures remain below

approximately -10 °C), such as the Ronne-Filchner and Ross ice shelves, albedo is higher during warmer summers compared to colder ones. This can be attributed to increased atmospheric moisture content during warm summers, leading to increased cloudiness and snowfall. Both fresh snowfall and increased cloudiness increase surface albedo. In RACMO2.4p1, albedo is



**Figure 4.** Spatial variability in the temperature–melt relationship over Antarctic ice shelves. Scatter plots show the relationship between summer melt and temperature from RACMO simulations, forced by ERA5 (1979–2023, purple), CESM2 (1985–2099, pink), and MPI-ESM (1985–2099, yellow), with grey background scatter representing all ice shelves and all RACMO simulations. Black line shows exponential fit between temperature and melt for the considered ice shelf, with dashed line showing the 97.5th percentile confidence interval. Boxplots of summer snowfall rates are shown for each simulation. The boxplots show the distribution of summer snowfall for each simulation, with the box representing the middle 50% and whiskers extending to values within 1.5× the interquartile range. The map displays the average surface melt for 2070–2099 in RACMO(CESM2) (shading), overlaid on a grey elevation map of Antarctica.

wavelength-dependent, allowing the model to capture the spectral effect of clouds on albedo. Cloud cover scatters much of the (near) infrared radiation, which has a low albedo, leaving mostly visible light, for which the spectral albedo is higher (van Dalum et al., 2020). This effect is illustrated in Fig. 6, which shows the difference in albedo between modelled (all-sky) and hypothetical clear-sky conditions across temperature bins. The figure demonstrates that, across all temperature ranges, warming is associated with increased cloud cover, which increases albedo. The cloud effect on albedo increases for lower albedos, as snow metamorphism primarily lowers the spectral albedo for red and near-infrared light, enhancing the cloud effect (Gardner






and Sharp, 2010).


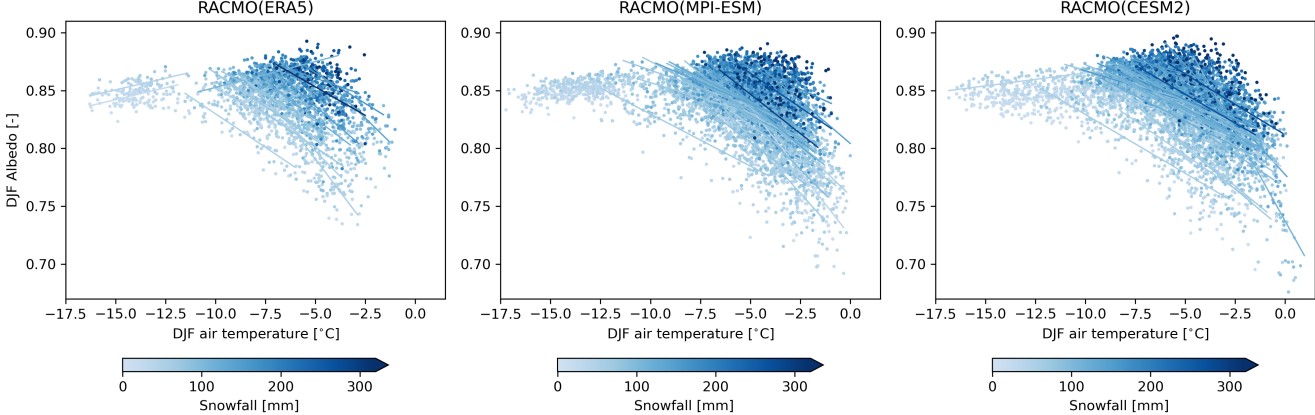

**Figure 5.** Temperature dependence of surface albedo and its relationship with snowfall rates in RACMO simulations forced by ERA5 (1979–2023) (a), CESM2 (1985–2099) (b), and MPI-ESM (1985–2099) (c). Each scatter point represents the average over one summer season (DJF) over one major Antarctic ice shelf (ice shelves with area > 800 km$^2$). Linear fits are shown per ice shelf where the R$^2$ value exceeds 0.3, with the color of each line indicating the average snowfall rate for that ice shelf.

On many of the warmer ice shelves (T$_{air, DJF}$ > -10 °C), cloudiness increases with temperature (not shown), similar to cold regions. However, its impact on albedo is outweighed by the effect of snow melt and snow metamorphism on albedo. While Fig. 6 shows that increased cloud cover can raise albedo by up to 0.05, the net albedo change on the warmer ice shelves in Fig. 5 is a decrease of larger magnitude. Antarctic snow albedo is largely dependent on the snow grain size, and the coarsening of

grains, snow metamorphism, happens at a rate that increases with temperature (Picard et al., 2012; Taillandier et al., 2007). Additionally, when snow melts and refreezes, the grain size increases, which enhances light absorption and further lowers albedo, a process known as the snowmelt–albedo feedback (Jakobs et al., 2019, 2021). Whether melt or dry metamorphism plays a greater role in lowering albedo depends on the the timing of snowfall relative to melt events. Besides, the magnitude of the albedo decline depends on snowfall rates (Fig. 5), as frequent snowfall brings small-grained, highly reflective snow to

the surface. This helps explain our previous findings that ice shelves that receive more snowfall have a lower melt sensitivity to near-surface air temperature compared to drier ice shelves. Frequent fresh snowfall dampens metamorphism and the snowmelt-albedo feedback, consistent with Jakobs et al. (2021).

## 3.5 Temperature dependency of SEB components

Next, we systematically assess how each SEB component responds to temperature and contributes to the non-linear relationship between temperature and melt. For each ice shelf grid cell that experiences an average summer melt of at least 1 mm





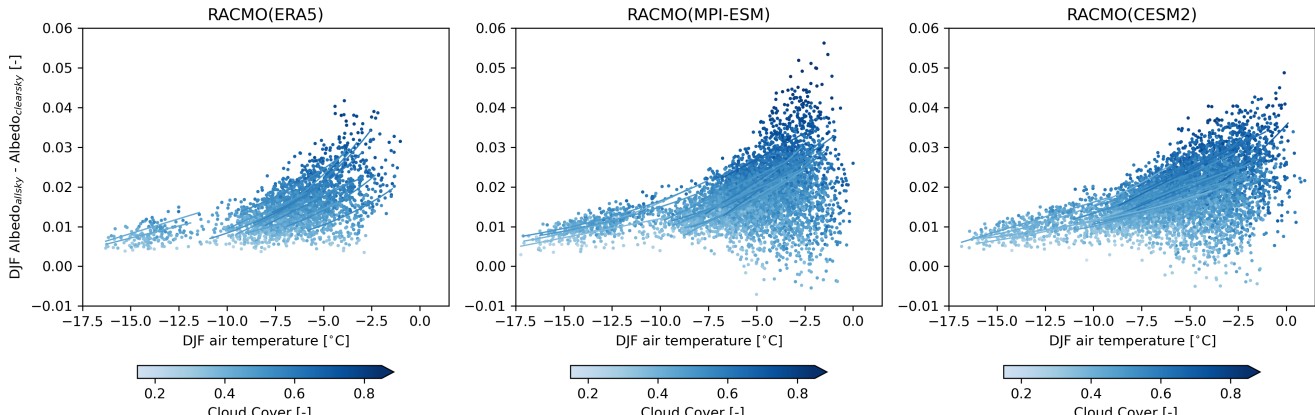

**Figure 6.** As in 5 but for the effect of clouds on albedo, with DJF cloud cover in color shading.

during the historical period, we fit either a linear or exponential function relating the summer average SEB component to air temperature. The function with the higher $R^2$ value is selected, and grid cells where the optimal fit parameters could not be reliably estimated (average $R^2 < 0.3$) are excluded from the analysis. We then calculate the average slope of the selected fits

across all grid cells, grouped into temperature bins. If the selected fit is exponential, the slope at the middle of the bin is taken. Figure 7 shows these average slopes for 'dry' ice shelves, defined as ice shelves with annual snowfall < 500 mm following van Wessem et al. (2023). The resulting slopes vary across temperature bins because each bin averages multiple fits with different temperature sensitivities, and the temperature range covered by fits at each location represents only a subset of the full temperature range. Uncertainty bands are shown based on the standard deviation between the fits (reflecting variability between locations) weighted with the average $R^2$ value (reflecting the strength of the fit). $R_{net}$ represents net radiation ($SW_{net} + LW_{net}$).



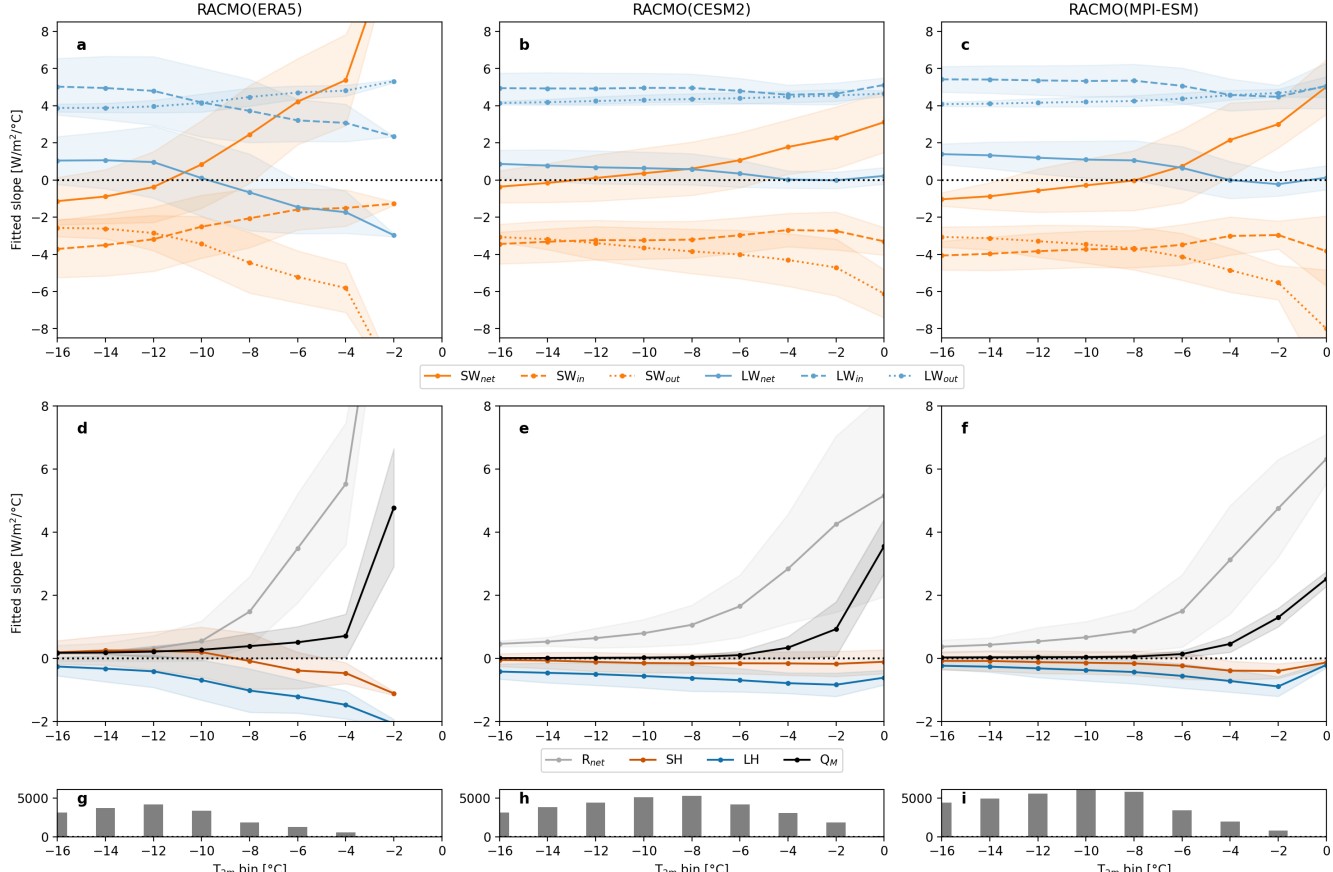

**Figure 7.** Temperature-dependent average fitted slope between summer surface energy balance (SEB) components and summer air temperature over dry ice shelves (annual precipitation < 500 mm). Each point represents the average slope of the SEB term with respect to temperature within a 2 °C bin, calculated over all dry ice-shelf grid cells that experience melt. Shaded areas indicate the weighted standard deviation, representing the error due to spatial variability across grid cells and strength of the fit. Columns represent the different RACMO simulations and the number of grid cells used to compute the average slope per bin is shown in g, h, i.

Net shortwave radiation ($SW_{net}$) responds differently to changes in temperature at low versus high temperatures (Fig. 7a,b,c). At lower temperature bins, $SW_{net}$ decreases due to a decrease in incoming shortwave radiation ($SW_{in}$) with rising temperature, caused by increased atmospheric moisture and cloud cover that reflect more sunlight. This decrease in $SW_{in}$ is larger
than the reduction in outgoing shortwave radiation ($SW_{out}$), because surface albedo increases with temperature in this range and $\delta SW_{net}/\delta T$ remains negative (Fig. 5). However, at higher temperature bins, this pattern reverses: $SW_{out}$ begins to decrease faster than $SW_{in}$ with warming as albedo drops due to snow metamorphism and melt, leading to $\delta SW_{net}/\delta T > 0$. Thus, on dry ice shelves net shortwave radiation ($SW_{net}$) first declines with warming and then increases with further warming, producing a distinct crossover point visible across all three model combinations. The exact temperature at which this crossover occurs
varies between –13°C and –8°C across the simulations, but it broadly aligns with the –11°C threshold where the strength of





the snowmelt–albedo feedback increases, peaking around –8°C as quantified by Jakobs et al. (2021).

Longwave radiation components exhibit strong and consistent temperature response. Outgoing longwave radiation (LW$_{out}$)
increases with near-surface air temperature due to its dependence on surface temperature, as governed by the Stefan-Boltzmann
law. Meanwhile, incoming longwave radiation (LW$_{in}$) increases too due to a warmer, moister atmosphere. This increase in LW$_{in}$
is generally stronger than the additional LW$_{out}$ from a warmer surface, resulting in a net gain in longwave radiation. In the simulation forced by ERA5, however, LW$_{out}$ begins to increase more rapidly than LW$_{in}$ at -10 °C (Fig. 7a), likely due to a weakening
in sensitivity of LW$_{in}$ and difference in cloud response to the other model forcings. Together, changes in net shortwave and net
longwave radiation nearly compensate for each other at lower temperatures, yielding a small effect on net radiation (R$_{net}$ in
Fig. 7d, e, f). But at higher temperatures, the snowmelt-albedo feedback drives a strong increase in net radiation, amplifying
melt.

The relative contributions of sensible and latent heat fluxes to the temperature response of the SEB are small compared to
that of the radiative components, as the turbulent fluxes are generally smaller components in the SEB during summer (e.g. Fig.
A1). At higher temperatures, the sensible heat flux decreases despite rising air temperatures, because the darkening of the snow
decreases the air-surface gradient due to increasing surface temperatures. The latent heat flux is on average negative in summer
(not shown), indicating net sublimation. This flux becomes increasingly negative with rising air temperature, meaning that
more energy is used for sublimation. Warmer air can hold more moisture and since the overlying air is typically dry due to its
inland/katabatic origin, the humidity gradient between surface and atmosphere increases. Additionally, the saturation specific
humidity at the ice surface increases exponentially with surface temperature, further increasing the vapor pressure gradient and
sublimation. The energy losses through turbulent fluxes compensate for energy gains from net radiation, but this only holds
at lower temperatures. At these low temperatures, most of the days included are non-melting days, and Q$_M$ changes little for
increasing temperature.

When considering wet ice shelves (defined by annual snowfall > 500 mm, following van Wessem et al. (2023)), which are
generally warmer, the picture looks different than for dry ice shelves (Fig. 8). For these wet ice shelves, fewer grid cells fall
into the lowest temperature bins, while a larger number of points contribute to the warmer bins. The response of net shortwave
radiation to warming is weaker and less consistent across the model forcings, which may result from differences in cloud and
precipitation patterns among the models in the coastal regions of the Amundsen Sea (Fig. 1), where many of these wet ice
shelves are located. The lower temperature sensitivity of net shortwave radiation over wet ice shelves is consistent with our
findings in Fig. 5, which show that snowfall dampens the snowmelt–albedo feedback. A notable difference from the dry ice
shelves is the response of net longwave radiation to warming. At higher temperature bins, net longwave radiation change with
warming remains positive on wet ice shelves, whereas it stabilizes or becomes negative on dry ice shelves. This shows the
influence of increased humidity and cloudiness that further increase incoming longwave radiation and contribute to melt on





wet ice shelves.

For both dry and wet ice shelves, several factors intensify melt rates when approaching summer air temperatures of 0 °C: first, air temperatures can exceed 0 °C while the surface remains at the melting point, allowing sensible heat flux to become positive and provide extra heat. Second, energy loss through sublimation levels off, since the saturation vapor pressure over

ice no longer increases when surface temperature halts at 0 °C, suppressing further sublimation while the air in the boundary layer can become more humid. Third, snowfall starts to transition to rain, therefore precipitation events can no longer raise the surface albedo and refreezing of rainwater can lower the albedo, leading to further increases in net shortwave radiation and net radiation. Rainfall begins to occur in the RACMO simulations at summer average air temperatures around –7.5 °C and increases to about 50 mm per summer by –2.5 C. Above -2.5 C, it starts to rise sharply, following an exponential trend (not

shown). These combined processes contribute to a rapid increase in melt on shelves near summer-average air temperatures of 0 °C.

**Figure 8.** As in 7 but for ice shelves in wet climates (annual precipitation > 500 mm)



## 4 Discussion and conclusions

Our study examines the spatial variability in the non-linear relationship between summer near-surface air temperature and sur-
face melt across Antarctic ice shelves, and aims to identify the key physical processes driving this non-linearity. To investigate
this, we used the regional climate model RACMO to dynamically downscale ERA5 reanalysis data for historical simulations
and two historical and future climate scenarios from ESM simulations. Building on Trusel et al. (2015), who related summer
near-surface air temperature to surface melt using an exponential fit, we extended the analysis by examining its spatial vari-
ability and physical drivers. We find that there is no single relation between temperature and melt, but that there are differences
in the exponential relationship between ice shelves in dry versus wet climates. For a given summer average air temperature,
dry ice shelves tend to experience more surface melt compared to wetter ones. While most ice shelves in drier climates are
currently still cold and relatively stable with little melt, these ice shelves could experience rapid increases in melt under future
warming. An example of a dry ice shelf that already experiences high melt rates is Amery Ice Shelf, where the largest amount
of surface meltwater is observed from sattelites in East Antarctica (Tuckett et al., 2025).


Several studies have proposed possible explanations for the non-linearity in melt sensitivity to temperature (Trusel et al.,
2015; van Wessem et al., 2023). However, this is the first study to systematically assess the relationship between summer air
temperature, surface energy balance (SEB) components, and their influence on surface melt in Antarctica. We find that the
non-linear relationship is strongly related to the temperature dependency of net shortwave radiation. At lower temperatures,
warming leads to increased cloudiness and snowfall, which reduce net shortwave radiation. However, at higher temperatures,
the metamorphism-albedo and snowmelt-albedo feedback become dominant, increasing net shortwave radiation, particularly
on dry ice shelves. This dominant role of net shortwave radiation is consistent with findings from Franco et al. (2013) on the
Greenland Ice Sheet, where albedo changes were found to be the most temperature-sensitive factor driving melt. However,
similar to Antarctica, the relative contribution of individual SEB components in Greenland also varies spatially. At the margins
of the Greenland Ice Sheet, the sensible heat flux exhibits strong temperature sensitivity and plays a major role in driving melt
anomalies (Braithwaite and Olesen, 1990; Franco et al., 2013; van den Broeke et al., 2008; Wang et al., 2021). In contrast, over
Antarctic ice shelves we find that the sensible heat flux is generally less sensitive to temperature, though it increases signifi-
cantly as summer air temperatures approach the melting point. This difference reflects the colder conditions over Antarctic ice
shelves compared to the Greenland margins where air temperatures are more frequently positive.


Once mean summer air temperatures reach the melting point, the processes governing surface melt begin to shift. Not only
the sensible heat flux becomes more important near the melting point when surface temperatures become more often con-
strained at 0 ° C, also the atmospheric temperatures and moisture content can continue to rise. As a result, energy losses
through sublimation decrease and net longwave radiation transfers extra energy to the surface for melt. At the same time, the
occurrence of rainfall increases. This transition from snowfall to rainfall reduces surface albedo and further accelerates melt
through a positive feedback. In these areas, rainfall can precondition the snow surface for enhanced melt (Nicolas et al., 2017;





Vignon et al., 2021), something already observed on the Greenland Ice Sheet (Doyle et al., 2015). In our simulations, annual rainfall averaged over the ice shelves increases by 4-7 mm over the 21st century, with the largest increase over the Antarctic Peninsula (15 - 40 mm increase) and ice shelves in West Antarctica (15-20 mm increase). However, there is a large spread

in projected rainfall changes across CMIP6 models (Vignon et al., 2021). Rainfall observations and high-resolution regional climate modeling are needed to improve rainfall simulation and to assess the role of rainfall in enhancing melt and affecting ice shelf stability (Gilbert et al., 2025).

Surface melt is strongly related to temperature because the SEB components themselves are temperature dependent. We

show that melt sensitivity to temperature varies spatially across Antarctic ice shelves, due to regional differences in snowfall, surface albedo and cloud cover, which influence the dominant SEB components and their sensitivity to temperature. These findings suggest that using a uniform, temperature-only approach, such as a fixed degree-day factor in positive degree day modeling, oversimplifies melt prediction and misses important regional differences and feedbacks. For example, ice shelves in dry climates tend to experience more melt at a given air temperature than ice shelves in wetter climates, partly due to

lower snowfall rates, which limits fresh snow accumulation, and enhances the snowmelt–albedo feedback. Therefore, spatially varying degree-day factors are needed to more accurately represent melt sensitivity in different climate regimes (Zheng et al., 2023). Alternatively, melt parameterizations could be improved by incorporating additional variables such as precipitation or albedo (Garbe et al., 2023), or by shifting toward fully resolved SEB formulations.

Our findings have important implications for understanding the vulnerability of Antarctic ice shelves to climate change. The higher melt rates for the same temperature on dry ice shelves suggests that these regions may be more sensitive to future warming than previously recognized, as earlier estimates did not account for regional differences in melt sensitivity. Not only their higher melt sensitivity, but also their lower snowfall rates mean they are more likely to reach the melt-over-accumulation (MOA) threshold of around 0.7 at which meltwater ponding and hydrofracture become possible (van Wessem et al., 2023;

Donat-Magnin et al., 2021). A recent study based on satellite observations confirmed that these drier ice shelves in East Antarctica are more favorable for meltwater ponding compared to ice shelves in West Antarctica (Tuckett et al., 2025). Our findings support and extend on the study by van Wessem et al. (2023) and Veldhuijsen et al. (2024), emphasizing an underestimated risk of ice shelves in dry climates to contribute to Antarctic mass loss.

*Data availability.* Monthly SMB components, SEB components and near-surface variables (e.g. temperature, windspeed, pressure) from RACMO2.4p1 simulations forced by ERA5, CESM2 and MPI-ESM are available at: https://doi.org/10.5281/zenodo.16902106



*Author contributions.* MGH led the conceptualization and analysis with guidance from WJvdB and MvdB. CvD and WJvdB developed this version of RACMO and, together with KV and MvT, performed the simulations and postprocessing. MGH prepared the manuscript with contributions from all co-authors.

*Competing interests.* One of the (co-)authors is a member of the editorial board of The Cryosphere. The authors have no other competing interests to declare.

*Acknowledgements.* This publication was supported by PolarRES. The PolarRES project received funding from the European Union's Horizon 2020 research and innovation programme call H2020-LC-CLA-2018-2019-2020 under grant agreement number 101003590. This research was also supported by Ocean Cryosphere Exchanges in ANtarctica: Impacts on Climate and the Earth system, OCEAN ICE, which is funded by the European Union, Horizon Europe Funding Programme for research and innovation under grant agreement Nr. 101060452, 10.3030/101060452. OCEAN ICE contribution number 35. We acknowledge the ECMWF for storage facilities and computational time on their supercomputer

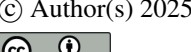


## Appendix A: Demonstration of the pseudo-SEB approach

To illustrate that radiation penetration considerably alters the SEB, Figure A1a compares the SEB over the Larsen C Ice Shelf
as simulated by RACMO2.3p2, which does not account for solar radiation penetration, with the SEB from RACMO2.4p1 (Eq.
(1)). Because of penetration of solar radiation in RACMO2.4p1, the net shortwave radiation at the skin layer is smaller, and
part of the radiation absorbed in the subsurface is resupplied as energy to the surface through $Q_G$. Because a large fraction of
the melt occurs now as internal melt, $Q_{M,\,surf}$ is smaller in RACMO2.4p1 compared to RACMO2.3p2. Figure A1b shows the
pseudo-SEB of RACMO2.4p1 (Eq. (2)), which is comparable to the SEB in RACMO2.3p2. $Q_M$ is here the energy used for
both surface and internal melt and agrees better with $Q_M$ in RACMO2.3p2, which demonstrates that the pseudo-SEB approach
is valid. Differences in the energy fluxes between the model versions also arise from differences in simulated near-surface
climate, albedo (between Dec-Mar) as well as considerable year round changes in cloud cover leading to less downwelling
longwave radiation (van Dalum et al., 2025), subsequently altering e.g. $LW_{net}$ and SH.

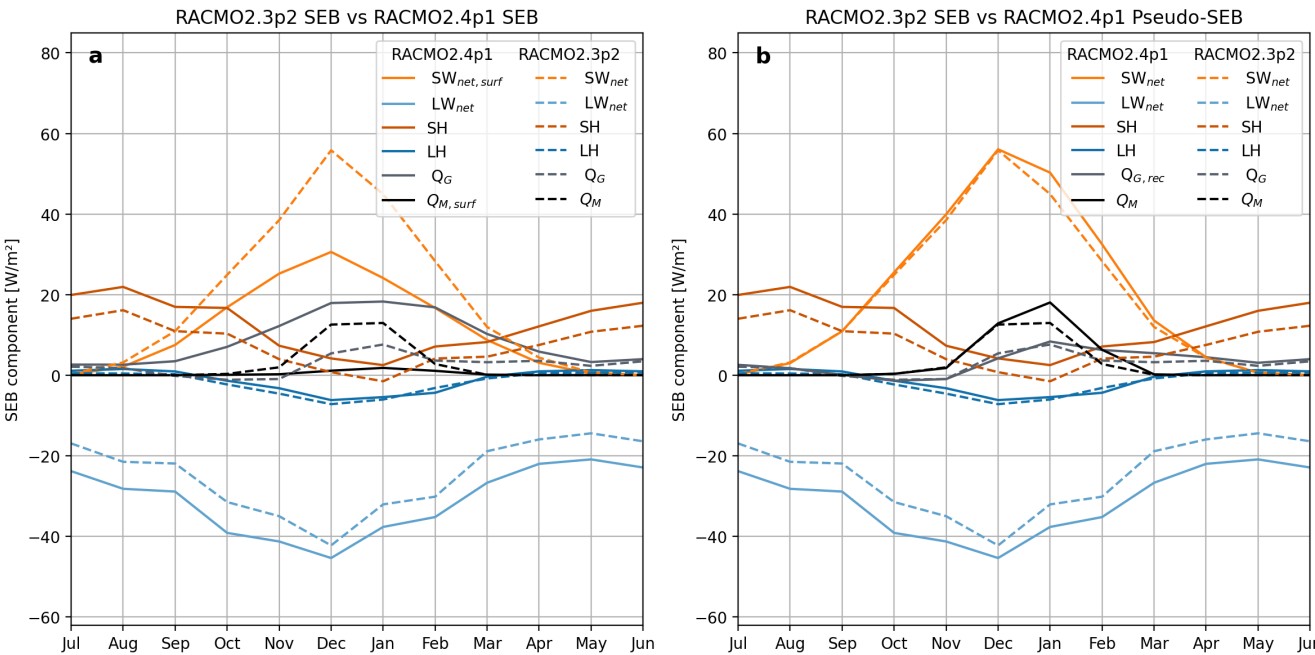

**Figure A1.** Monthly mean surface energy balance (SEB) components from RACMO2.4p1 and RACMO2.3p2 for 2001–2016, spatially
averaged over the Larsen C Ice Shelf. In a) the SEB from RACMO2.4p1 (solid lines) is given as in Eq. (1) and in b) the pseudo-SEB for
RACMO2.4p1 (solid lines) is given as in Eq. (2). In both panels, RACMO2.3p2 estimates (dashed lines) are shown for comparison. The data
are taken from van Dalum et al. (2024), as these simulations are on the exact same grid.





**Appendix B: Evaluation of RACMO historical simulations with ESM forcings for summer**

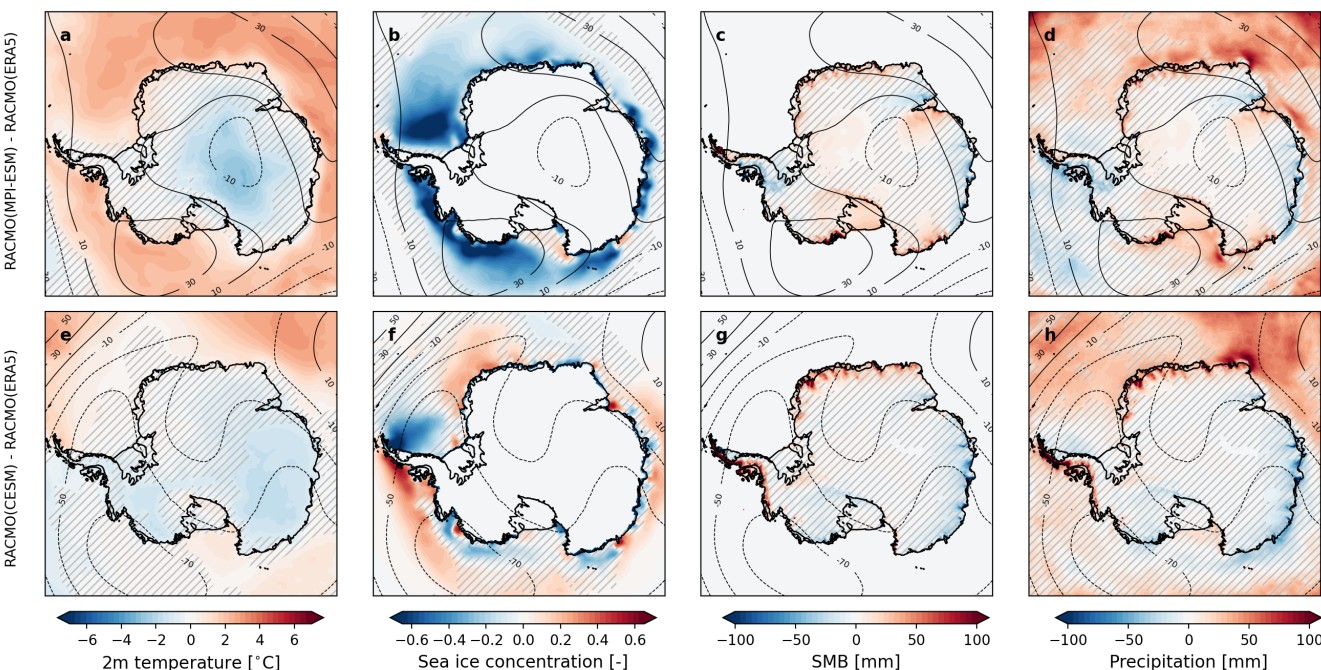

**Figure B1.** As in Fig. 1 but for DJF.



## Appendix C: Exponential relationship temperature-melt and snowfall

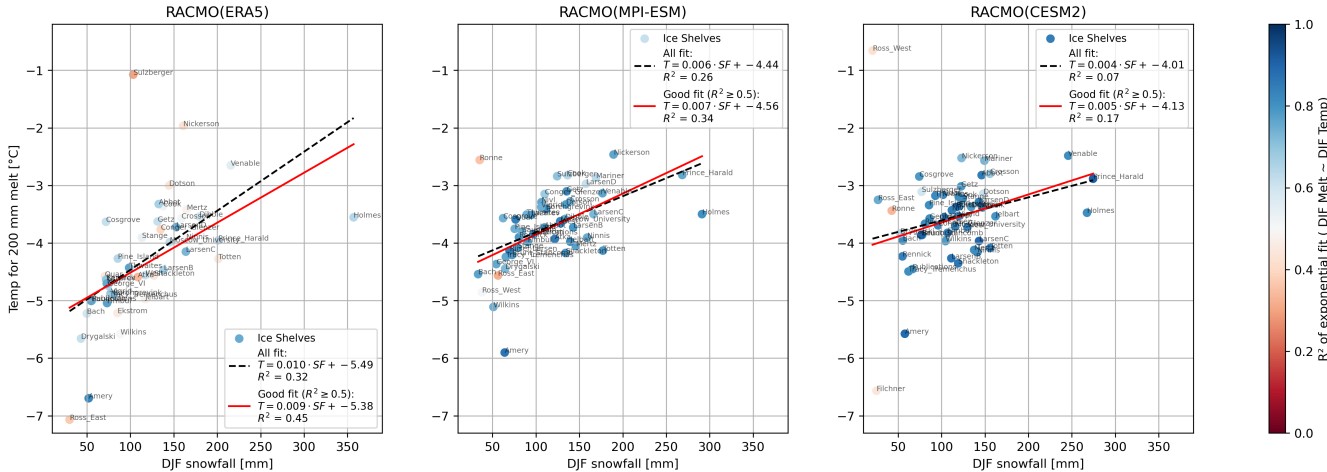

**Figure C1.** Average DJF snowfall rates over the historical period vs. DJF near-surface air temperature required to produce 200 mm of surface melt according to fitted exponential relationships (Fig. 4) for major Antarctic ice shelves, across the three RACMO simulations (columns). Each point represents one ice shelf, colored by the $R^2$ of an exponential fit between melt and temperature. Two linear regression lines are shown: a black dashed line fit to all data points, and a red solid line fit only to shelves where the exponential fit had a $R^2 \geq 0.5$. These lines illustrate the relationship between snowfall and the temperature sensitivity of surface melt.





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
