# Peer review of "On the non-linear response of Antarctic ice shelf surface melt to warming"

_EGUsphere, 2025_

## Author Comment (AC1)

**Author response to Referee #1**

This manuscript provided a spatially informed comparison of the mechanisms which dictate differences in the relationship between rising air temperature and meltwater production over the ice shelves of Antarctica. The results represent a novel and potentially important contribution to the literature pending suitable revisions. I have included some general feedback followed by a list of specific recommendations below.

We would like to thank the referee for their comments and we address them here. In black are given the comments, in blue our response and in orange the changes we would implement in the manuscript.

General comments

The authors briefly state that RACMO may struggle to accurately represent longwave radiation and turbulent heat fluxes in the opening paragraph of section 3.1. This deserves more discussion. How does RACMO struggle to represent these variables? Are the biases in these variables uniform across antarctica or regionally dependent? How does RACMO accurately resolve surface air temperature and SMB if these important surface energy balance components are not resolved accurately in the model? What implications, if any, do these shortcomings have for the results of the study?

We plan to given more detail on the biases in section 3, including the locations of the weather stations where these biases are based on. We also will briefly discuss how the biases in SEB terms translate into melt estimates as follows (note that this text will be moved to the methods section 2.3 as suggested by Referee #2):

"RACMO(ERA5) has been extensively evaluated against weather station and mass balance observations in van Dalum et al. (2025). Using observations from automatic weather stations on the Antarctic Peninsula and in Dronning Maud Land, this study shows that version 2.4p1 performs well in simulating Antarctica's near-surface air temperature (bias of -1.40 and an RMSE of 4.38 °C) and shortwave radiation (bias of 8.5 and -8.8 W m−2 for downward and upward shortwave radiation, respectively), but has larger differences with observations for longwave radiative fluxes (bias of -20.4 and 11.7 W m−2 for downward and upward longwave radiation, respectively). Turbulent fluxes have small bias (-0.3 and 1.5 W m−2 for latent and sensible heat flux, respectively), but large spread (RMSE of 5.0 and 14.2 W m−2, respectively). As the biases in longwave and shortwave radiation partially offset one another, the resulting melt rates are less affected. This is reflected in the good agreement between simulated meltwater presence in the snow and satellite-based estimates (van Dalum et al., 2025)."

One of the strengths of this study is the detailed consideration in climate-driven spatial differences in SMB response between the various ice shelves. Some of the results presented in section 3.1 worked counter to this strength. For example, spatial averages in Table 1 could obscure biases that are locally relevant to a specific ice shelf. Furthermore, I did not see how this examination of biases was considered in the interpretation of rest of the results. Do these biases have any implications for the

conclusions of the study? Here again, spatial averaging makes it difficult to answer this question.

Thank you for this comment, we agree that the discussion of biases in the ESM-forced simulations would benefit from more discussion on regional differences. Therefore we have now added an extra figure with difference plots, as in Fig. 1, but for the main SEB terms (Swnet, Lwnet, LH, SH), where we can discuss in more detail the regional differences between the ESM-forced and ERA5-forced simulations. The results of Table 1 are now less central in the text and that is why it is moved to the appendix.

Note that now also the difference maps in the main text display the differences for DJF, to have more consistent focus on summer air temperatures and SEB. We have replaced the sub-figure for SMB with melt in Fig. 1 to be able to discuss the regional differences in the biases in melt.

That the differences between ESM- and ERA5-forced simulations did not impact our results is added to the discussion: "We find that the temperature sensitivity of the surface energy balance components are very similar across all simulations, indicating that the ESM-forced simulations reliably reproduce the relevant physical processes and can therefore be used to extend the temperature and melt range beyond that from the historical RACMO(ERA5) simulation."

I often found it difficult to see from where the authors were basing their claims. I believe the manuscript would benefit from more detailed explanations of how the figures support their claims. This is particularly true for the discussion of figures 4-6.

In response, we have clarified the description of Figure 4 and moved Appendix Figure C1 to the main text, as it provides more quantitative evidence for the spatial variability in the temperature-melt relationship linked to snowfall. In addition, we have improved the interpretability of Figures 5 and 6 by changing the trend lines to black and labeling selected lines. This allows for a clearer reference to the relationships shown in these figures.

Specific comments

L13: delete the s from "becomes"

Done

L74: I would suggest rewriting as "...and therefore provide a better representation of areas such as..."

Done

L79: delete "in" from "penetrate in the snowpack"

Done

L103: It can be assumed from the description that ER>0 for erosion and ER<0 for deposition, but it would not hurt to state this explicitly.

We have included this.

L109: A citation is needed for the ERA5 reanalysis dataset

We have included this now: "We use the RACMO simulation forced by ERA5 reanalysis data (Hersbach et al., 2017),.."

L111: "Future projection simulations" is a bit redundant. I would suggest "Projections spanning 2015 to 2099 were forced using…"

Thank you, we have now changed this.

L114: A short rationale for why SSP3-7.0 was chosen could be included here.

We clarify this as follows: "SSP3-7.0 was chosen within the PolarRES framework because it is considered a more plausible high-emission pathway than SSP5-8.5."

L117-120: This information would fit better in the previous paragraph, which is where you first introduce the future period simulations.

Thank you for the suggestion, we will restructure so that the first paragraph discusses the projection runs as part of PolarRES, and the second paragraph discusses the ERA5 forced simulation as reference simulation. The first pagraph then reads as follows:

"We use RACMO2.4p1 with a domain covering Antarctica and the southern tip of South America with a horizontal resolution of 11 km, forced with both reanalysis and Earth-System-Model (ESM) datasets. The simulations for this domain were performed as part of the PolarRES project, an EU Horizon 2020 funded project that uses RCMs to simulate the current and future climate of the polar regions. Projections spanning 2015 to 2099 were forced using boundary conditions from two CMIP6 ESMs: the Community Earth System Model 2 (CESM2) and the Max-Planck Institute Earth System model (MPI-ESM), under the high emission SSP3-7.0 scenario. SSP3-7.0 was chosen within the PolarRES framework because it is considered a more plausible high-emission pathway than SSP5-8.5. CESM2 and MPI-ESM were then selected from the CMIP6 ESMs using a storyline approach to represent two contrasting but plausible Antarctic climate futures: CESM2 reflects a future with extensive sea ice loss and an earlier summertime stratospheric polar vortex breakdown, while MPI-ESM captures a scenario with limited sea ice loss and a delayed polar vortex breakdown (Williams et al., 2024).

L124: Is this calculation of sea ice temperature performed within RACMO? Also, a citation is needed for this slab model.

We will clarify that the sea ice slab model is part of the ECMWF IFS model that was cited before in the text. "Sea ice temperature is calculated using the four-layer sea ice slab model from the ECMWF IFS model, which assumes a fixed maximum thickness of 1.5 m."

L130: Can you clarify what is meant by "average climate in the historical simulation..."? As written, I would assume that RACMO(ERA5) is the historical simulation, since it is forced by an observationally constrained dataset. From what is written in the remainder of the paragraph, it seems the comparison referenced here is between RACMO(ERA5) and RACMO forced by the ESM's representation of historical conditions. It is also not clear what the "average climate" is here. Perhaps long-term mean would be more accurate?

To clarify this we rephrase this sentence to: "Because ESM-forced simulations are not constrained by data assimilation, we evaluate the ESM-forced simulations by comparing the mean and variability of near-surface climate variables over 1985–2014 with those from RACMO(ERA5) for the same period."

L132: Is this statement summarizing your own attempts at validating RACMO against in situ data? If so, where is this data presented? If not, a citation is needed.

This statement is based on the paper by van Dalum 2025, which is clarified as follows "RACMO(ERA5) has been extensively evaluated against weather station and mass balance observations in van Dalum et al. (2025). Using observations from stations on the Antarctic Peninsula and in Dronning Maud Land, this study shows that version 2.4p1 performs well in simulating Antarctica's near-surface air temperature (bias of -1.40 and an RMSE of 4.38 ∘C) and shortwave ..."

L138: The word "significant" is usually reserved for instances of statistical significance. Was a statistical analysis performed here? If so, what method was used and where are these results presented?

We have not done a statistical test and therefore will remove the wording 'significant'. Instead we compare the difference with the typical year-to-year variability through comparing it with the standard deviation. We have rephrased L138 to: "Over the Antarctic continent, RACMO(MPI-ESM) temperatures differ little from RACMO(ERA5) relative to the year-to-year variability. Exceptions are Dronning Maud Land, which is warmer, and the high Antarctic Plateau, where temperatures are lower than RACMO(ERA5) by more than the inter-annual standard deviation."

Figure 1: It is somewhat unusual in my experience to use hatching to highlight areas of small differences relative to internal variability. This is also a bit confusing in the context of the discussion, where significant differences are emphasized (L138). This is related to my previous comment, but if a statistical significance test was conducted, I think it would make more sense to highlight areas of statistical significance.

In combination with the previous comment, we now avoid use of the word 'significant' when referring to the difference between simulations, but rather discuss in terms of larger/smaller than inter-annual standard deviation. We also have changed the hatching so it indicates regions where the difference is larger than the inter-annual standard deviation. We have clarified the use of hatching in the figure in the text by adding the following: "In the figure, hatching indicates areas where the mean difference between the ESM-forced simulations and RACMO(ERA5) is larger than the ERA5 interannual standard deviation over the historical period."

Table 1: If the focus of this paper is on the Antarctic ice shelves, how relevant is a spatial average of model biases across the whole of the Antarctic Ice Sheet? It seems that spatially informed biases are of critical importance to the question at hand, and the information in Table 1 may mask some of these locally relevant biases by averaging biases of opposing sign in different regions (e.g., the strong negative precipitation bias over east Antarctica versus the strong positive bias over Dronning Maud Land in RACMO(MPI-ESM)).

We agree that spatially averaged biases can mask locally relevant differences, especially as there are regions with opposing signs in biases. We have now moved Table 1 to the appendix and instead added spatial maps of the differences in main SEB terms (SWnet, Lwnet, LH and SH) between the ESM-forced simulations and RACMO(ERA5). The text is therefore also discussing more the regional biases instead of Antarctic-wide integrated differences.

Figure 2: Panels are referred to by letter in the figure caption, but there is no lettering on the figure.

We have now included the lettering in the figure.

L185: "consistently stronger" is a bit unclear. Perhaps something like "...21st century; however, output from RACMO(CESM2) consistently shows a greater rate of warming than RACMO(MPI-ESM).

Thank you for this suggestion; we have now implemented this change.

Figure 3: The error bars in each panel can be hard to read. Would it be possible to spread them out more so as to avoid overlap?

We have now adjusted this.

L221: What are the authors relying on to make this claim about increased snowfall over cold ice shelves? This explanation makes sense from a physical standpoint, but did the authors verify an increasing trend in snowfall over these ice shelves in their RACMO simulations?

We have verified this by checking similar scatter plots between DJF air temperatures and DJF precipitation and cloud cover. The relation with cloud cover was most significant. We have rephrased the sentence to:
"This can be attributed to increased atmospheric moisture content during warm summers, leading primarily to  increased cloudiness, with a smaller contribution from increased snowfall (not shown). "

L224: Why is "near" in parentheses?

This should be near-infrared instead of (near) infrared. We have changed this now.

Figure 5 caption: Caption refers to figure panels by letter, but letter labels are missing from the figure.

Figure includes subplot labels now.

Figure 5: Points and lines are color coded according to average snowfall rate. I do not see where mean snowfall rate is discussed in the context of the albedo-temperature relationship.

This is discussed around line 240 in the original MS. We have added further discussion on this with reference to example ice shelves that are now indicated in the figure as follows:
"This is evident in Fig. 6 where the strongest decreases in albedo with increasing temperature occur at ice shelves with very low snowfall rates (e.g. Nansen and Publications ice shelves), whereas some ice shelves with high snowfall rates show little to no decline, or even an increase, in albedo."

Figure 5: it is difficult to read these plots. As noted by the author in the previous paragraph, one of the more interesting pieces of information conveyed here is the slope of the albedo-temperature relationship is different among ice shelves. This is evident in the fit lines, but it is hard to distinguish the fit lines from the points. Perhaps using different color scales for the fit lines and points could help? Also, while it is not practical to label all fit lines, perhaps annotating a few lines to highlight the difference between the relatively cold and warm ice shelves could clarify things.

We will adapt the figure so that the trend lines are not color-coded but are plotted in black on top of the scatter to improve clarity of the figure. We will add labels of the ice shelves to a selection of the fit lines.

Figure 7 caption: delete "is" from last line.

Done

L324: should be spelled "satellites"

Done

L341: Might read better as "Not only does the sensible heat flux become  more important … at 0 °C, but atmospheric temperatures and moisture content can also continue to rise."

Thank you for the suggestion, we have incoorporated this.

---

## Author Comment (AC2)

**Author response to Referee #2**

Review of "On the non-linear response of Antarctic ice shelf surface melt to Warming"

This study investigates non-linearity in the temperature-melt relationship over Antarctic ice shelves using a modeling approach. The goals of the study are twofold, first to investigate spatial variability in the temperature-melt relationship and, second, to identify the dominant components driving non-linearity in the temperature-melt relationship. The authors pursue both goals by analyzing outputs from a regional climate model forced over the historical period (1979-2023) using ERA5 and two future climate scenarios (SSP3-7.0) using CESM2 and MPI-ESM (2015-2099). I think the study addresses an important research topic that is certainly within the scope of The Cryosphere and it is generally presented in clear, concise way. However, I do have some concerns about the scientific rigor and significance which I think would require some fairly substantiative changes to address. My recommendation is that the manuscript is considered after major revisions.

General comments

1. It's not clear how Section 3.1 and 3.2 contribute to the goals of the study. I understand that the authors need some temperature variation to investigate its relationship with melt. But it looks like there is plenty of variation in the historical forcings (e.g. Fig. 5a and 6a). The authors should consider removing these sections or strengthening the links between ESM forcing/trends and the goals of the study.

Section 3.1 is included to get confidence in the performance of RACMO forced with ESMs before using these in the analysis. The trends and variability in temperature and melt in the future simulations in Section 3.2 form the basis of the analysis in the temperature/melt relationship, and therefore we critically assess these underlying data here. To make this clearer for the reader, we add a brief overview at the start of the Results section outlining what to expect:

'First, the performance of RACMO forced by the ESMs is evaluated to gain confidence in in the simulations before using them in the analysis. We then assess the temperature and melt trends in the future projection runs, which provide the basis for studying the temperature–melt relationship. We analyse the spatial variability in the relationship between temperature and melt, the role of albedo feedbacks in the non-linearity and systematically assess how all SEB terms depend on temperature and contribute to the melt response.'

2. On a related note, it's not clear why two different ESMs are used to force RACMO. The current use of two ESMs makes the figures cluttered (e.g. Figures 5 and 6) and distracts from the main message. The authors acknowledge that the relationship between temperature and melt is consistent across the ERA5 and ESM model forcings (L201-202). I recommend that the authors provide better motivation for using two models or consider streamlining the analysis by not using any (see previous comment) or just one.

Including the ESM-forced simulations helps to answer our research question because regions that experience little melt in the current climate experience more melt and variability towards the end of the 21st century (eg Filchner-Ronne Ice Shelf, Fig 3). Therefore the temperature-melt relationship can be analysed for more ice shelves, and reaching higher summer temperatures. For example, the RACMO(ERA5) simulation lacked data in the temperature bin around 0 °C, whereas the CESM2- and MPI-ESM-forced simulations do provide data in this range (Fig. 7). Including multiple ESM forcings and showing the consistency among them strenghtens our findings.

We added this motivation at the start of Section 2.3:
"Using ERA5 provides present-day conditions, while the ESM-forced simulations extend the analysis into future climates, allowing the temperature-melt relationship to be examined across a wider range of melt intensities and summer temperatures than occur today."

3. The authors categorize ice shelves in several ways. In Figure 3 the ice shelves are categorized regionally. In Figures 7 and 8, ice shelves are categorized by annual precipitation. In the text, the authors state that the ice shelves are categorized by summer air temperatures as well. The central message of the manuscript would be strengthened if the analysis categorized ice shelves in just one way.

We decided to keep the temperature and melt trends for different regions in Figure 3 because it serves two purposes: 1) it allows for evaluation of regional melt trends and variability and 2) highlights differences between ERA5- and ESM-forced simulations which would not be visible if ice shelves were already classified only as dry or wet. The dry/wet classification is introduced only after Section 3.4, once we demonstrate that the temperature-melt relationship differs between ice shelves in dry and warmer climates and provide an initial physical explanation based on albedo feedbacks. From that point onward, the analysis focuses on the wet/dry categorization to examine the underlying processes in more detail.

We have clarified this methodology by adding a section in the Methods (see following comment). From your comment we realise that the wording 'cold ice shelves' and 'warm ice shelves' in Section 3.5 could be interpreted as an additional classification. These terms referred to the lower and higher temperature bins used in the SEB-temperature slope analysis, and we have reworded them to lower and higher temperature bins to avoid confusion.

4. The motivation for categorizing ice shelves by annual precipitation is not clear. As written, it feels like the authors tuned an annual precipitation threshold until the data were split according to their narrative. I recommend that the authors better explain their decision to use a 500 mm threshold in both introduction and the methods sections. Better would be to provide a sensitivity analysis on this threshold.

We have now clarified this categorization in the Methods section as follows:
"Albedo and cloud feedback's are expected to play a dominant role, and therefore the analysis is split between iceshelves in relatively dry and wet climates, based on a

threshold in annual snowfall. Following van Wessem et al. (2023), we use the median annual snowfall of 500 mm yr−1 as the threshold separating 'dry' and 'wet' ice shelves, which results in two equally sized groups."

5. The authors claim that there is large spatial variability in the temperature-melt relationship on ice shelves but the current analysis is not very convincing. I acknowledge that there is some qualitative support for this finding in Figure 4. But the analysis would be improved by demonstrating this variability quantitatively or statistically.

Thank you for your comment. We agree that Fig 4 illustrates our statement, but does not provide quantitative support, therefore following your later suggestion at L209, we have moved Fig C1 to the main text. We have adapted the second paragraph of Section 3.3 to the following:
"Ice shelves in drier climates tend to experience more melt at the same temperatures compared those in wetter climates. For example, the Amery ice shelf, which receives less than 100 mm snowfall during summer (boxplot in Fig. 5), has a temperature-melt curve where melt start increases at much lower temperatures compared to other ice shelves (grey scatter Fig. 5). In contrast, the Nickerson Ice Shelf, located in a much wetter climate on the Marie Byrd Land coast, shows melt rates increasing only at substantially higher summer air temperatures. These examples show that the temperature required to reach a certain melt rate varies between ice shelves.

To further quantify this, we use the fitted exponential temperature-melt relationships and determine for each ice shelf the summer air temperature at which 200 mm of melt would occur. This temperature is plotted against summer snowfall rates in Fig 6. Although the slope and R2 of the linear regression vary between the simulations, all show a clear pattern: ice shelves in drier climates reach 200 mm of melt at summer air temperatures several degrees lower than those in wetter climates."

6. Too much of the main text and figures are focused on SEB components at very low temperatures when there is little or no melt. There is therefore a mismatch between the goals of the study and the findings. The manuscript would be improved by expanding Section 3.3. and focusing Sections 3.4. and 3.5 (and Figures 5-8) on air temperatures that are associated with melt. Alternatively, given that some of the most interesting findings occur at low air temperatures, perhaps the goal of the study could be modified (e.g. impact of air temperature on SEB).

We have revised these sections to better emphasize temperature ranges associated with melt. In Sections 3.4 and 3.5, we have strengthened the focus on higher temperature bins where melt occurs, for example by adding more discussion on SWnet curve at higher temperature bins for dry ice shelves: "The slope between SWnet and temperature continues to increase beyond this point in all simulations, initially driven by snowmelt refreezing and dry snow metamorphism. At the warmest temperature bins, albedo can decrease further due to the refreezing of rainfall and the increasing exposure of bare ice."

Similar for wet ice shelves: "The slope of SWnet is relatively constant across the temperature bins, except for the bin around 0C, where SWnet increases more rapidly with warming. In this temperature range, a fraction of the precipitation falls as rain, which removes the damping effect of fresh snowfall and reduces albedo through refreezing of rainwater."

The discussion of turbulent heat fluxes already focused on higher temperature bins, as it explicitly describes how sensible and latent heat fluxes change sign or level off in the highest bins, thereby contributing additional energy for melt alongside the increase in net radiation. Lastly, in the Discussion and conclusions section we have now emphasized more on conclusions about contribution of SEB sensitivity to melt sensitivity.

Specific comments

L24-25: It would be useful to add a brief statement here to explain why the relationship between temperature and melt is highly non-linear.

More explanation for this is given later in the introduction and by the extra discussion given in the following comment.

L47-49: The problem statement should be expanded by reviewing the findings of Jakobs et al. (2019) in a bit more detail and clarifying how this study differs from van Wessem et al. (2023). One way the authors could do this is by briefly reviewing how SEB components (e.g. albedo, clouds) are thought to respond to temperature in Antarctica.

We have split the last paragraph of the introduction in two and have expanded on the problem statement as follows: "One of the possible explanations for this non-linearity is the snowmelt-albedo feedback (Jakobs et al., 2019, 2021) in which melt and subsequent refreezing lowers the albedo of snow, increasing absorption of solar radiation and melt. The potential of this feedback to enhance surface melt is modulated by the frequency and timing of snowfall events in summer. But as noted by Jakobs et al. (2019), under warmer conditions such as those currently observed on the Antarctic Peninsula, the snow-melt albedo feedback becomes less important in enhancing melt, and other processes such as exposure of bare ice or turbulent fluxes play a role. In addition, other surface energy balance terms such as longwave radiation and the latent heat flux may also respond non-linearly to air temperature through changes in atmospheric moisture and cloud conditions that are associated with the atmospheric warming."

L65-66: Recommend placing the "van Dalum" reference before "are" and after "RACMO2.3p2"

We placed the reference instead in the previous sentence where the new version is mentioned for the first time: "In this study, we used the latest RACMO version 2.4p1 (van Dalum et al., 2024), in this paper referred to as RACMO. The main differences between RACMO2.4p1 and the previous operational version, RACMO2.3p2, are:"

L73: "part of" is redundant here if "grid cells can be partially glaciated…"

We have removed 'part of'.

L74: "…and therefore better representing areas…" is a little awkward, consider revising.

This is now rephrased to: "and therefore provide a better representation of areas such as the McMurdo Dry Valleys"

L82: Should it not be "SEB of skin layer is defined as:"?

We have included this now.

L92: energy "available" for melt since some will be used for warming

We have rephrased to "energy available for melt".

Section 2.2: It's not clear to me why SMB is defined here since it is not required to complete the goals of the study. Given the importance of melt in the study, it would be more useful to describe the relationship between QM and melt, instead of SMB.

We have included the SMB definition here because the precipitation component is strongly linked with the SEB and melt in this study, through its effect on albedo. We have added the following in the Introduction to make more clear the role that precipitation is expected to play for surface melt:
".. One of the possible explanations for this non-linearity is the snowmelt-albedo feedback (Jakobs et. al, 2019) in which melt and subsequent refreezing lowers the albedo of snow, increasing absorption of solar radiation and melt. The potential of this feedback to enhance surface melt is modulated by the frequency and timing of snowfall events in summer."

In the methods we have made the link more clear between melt and SMB, including how melt mass rate is calculated from QM:

'The resulting melt rate (kg m−2) is then obtained by converting the available melt energy into a meltwater mass flux using the latent heat of fusion, such that the energy remaining after warming the snow to the melting point produces melt. Melt contributes to mass loss when the resulting meltwater is not refrozen or retained in the snow and instead leads to runoff (RU). The surface mass balance (SMB) describes the balance between accumulation and ablation in the near-surface firn or ice:
SMB = Ptot − SUs − SUds − RU − ER (3)
with Ptot total precipitation, SUs is surface sublimation and SUds sublimation of blowing snow, RU runoff and ER drifting snow erosion which can be both ablative (erosion, ER > 0) and accumulative (deposition, ER < 0).'

L107-108: "have been done" is a little awkward, consider revising.

We have revised this to "The simulations for this domain were performed as part of the PolarRES project"

L129-134: Recommend moving this text to the Methods

We have now moved this section to the Methods, and more discussion is given there on the biases as suggested by reviewer #1.

L143: How was the anomaly computed? Recommend that this is described in the Methods

We have replaced 'anomaly' by 'difference' to clarify that we refer to the difference between simulation forced by ERA5 and by the ESMs. This is further clarified in the methods as follows:
"As a second evaluation step, historical simulations (1985-2014) forced with ESMs are compared to RACMO(ERA5) in Section 3.1. We evaluate whether the mean difference between the ESM-forced simulations and RACMO(ERA5) exceed the inter-annual standard deviation of RACMO(ERA5)."

L151-152: Recommend moving this text to the Methods

We have adapted the difference maps here so they show the difference in DJF mean instead of annual mean. This is done to more consistently show results focussed on the summer season throughout the manuscript. Therefore this statement is removed here.

L158-159: Not sure that Table 1 supports this statement.

We have now moved Table 1 to the Appendix and expanded the discussion of differences between the ESM-forced simulations and the ERA5-forced simulation by adding difference maps for melt and SEB terms. We have therefore replaced the original statement with a more nuanced statement of the overall performance:
"Overall, the ESM-forced simulations reproduce Antarctic near-surface climate, SEB, and melt patterns well over the ice shelves, with differences relative to RACMO(ERA5) generally within the inter-annual standard deviation, except in a few localized regions such as the Wilkins and George VI Ice Shelves."

L180: "trends of" near-surface air temperature?

We see that L180-183 are confusing here and give repeated information about the difference in temperature in the historical period that are already discussed with the difference maps. Therefore these sentences are removed here.

L181: Recommend that "inter-annual variability" is quantified to support this statement.

Not relevant anymore, see previous comment.

Fig. 3: "COLD" ice shelf category has not been defined yet, recommend doing so in the Methods

We have written out the abbreviations in full in this figure, making clear that "COLD" and the other abbreviations refer to a regional grouping and is not intended as an additional ice-shelf category.

L207: Not clear how drier and wetter climates are defined. Recommend describing in the Methods. Also see general comment.

This is now defined in the Methods section, see reponse to general comment 4.

L209-214: Reasoning is unclear – does Nickerson Ice Shelf also have a mean summer air temperature of -4C? It would be better is the study could quantitatively demonstrate this point instead of cherry-picking examples. I think Figure C1 could serve this purpose, consider adding it to the main text.

We have adapted this section in line with general comment 5. We have reworded this sentence to first give Amery Ice Shelf and Nickerson as examples of contrasting temperature-melt relationships in a wet versus dry climate. After this, we quantitatively demonstrate the link with snowfall using Fig C1. See general comment 5 for the proposed revised text.

L211-212: Awkward phrasing, consider revising.

Revised to 'These examples show that the temperature required to reach a certain melt rate varies between ice shelves.'

L212-213: Not clear what this sentence is getting at.

The revised text is given in our response to general comment 5.

L223: Recommend moving to Methods

Done

L226: How were these hypothetical clear-sky conditions derived? Was this a separate model run or something that is produced implicitly by RACMO. The methods could be improved by outlining the key metrics that are used for the analysis.

In RACMO, the shortwave radiation profiles are calculated separately for the case that the sky would be clear sky, and for a full overcast case. The clear-sky radiation fluxes are stored in this process. Then depending on the cloud cover, the actual shortwave radiation fluxes are calculated as weighted average based on cloud cover. We have explained this in the new methods section (2.4) as follows: "The influence of clouds on albedo is examined by comparing clear-sky and all-sky albedo, making use of the fact that RACMO separately calculates radiative fluxes under clear-sky and cloudy-sky

conditions and subsequently combines them into total-sky fluxes based on cloud fraction."

L248-255: Recommend moving to Methods

We will move this to the methods under a new section "2.4 Assessing temperature-dependency of SEB"

L258: Should "SWnet" actually b "the correlation between SWnet and air temperature"?

We have clarified this text as follows:

The relationship between net shortwave radiation (SWnet) and temperature is not constant across temperature bins (Fig. 8a,b,c). In the lower temperature bins, the slope of SWnet is negative, indicating that SWnet decreases with warming. This decrease occurs because higher atmospheric moisture and increased cloud cover reduce incoming shortwave radiation (SWin) by reflecting more sunlight. This decrease in SWin is larger than the reduction in outgoing shortwave radiation (SWout), because surface albedo increases with temperature in this range and $\partial SWnet/ \partial T$ remains negative (Fig. 6).

L267: "peaking" seems to imply that the strength of the snowmelt-albedo feedback reduces at temperatures higher than -8 but Fig. 7 does not show that. Please consider clarifying.

Thank you for this comment. We agree that the previous wording was misleading. In Jakobs et al., the "strength" of the snowmelt–albedo feedback refers to the contribution of this feedback to total melt. Thus, a "peak" in their study indicates that, beyond this point, other processes become increasingly important in driving melt, even though the albedo itself can continue to decrease. To avoid confusion, we have removed this reference from the main text here and instead discuss it in the Discussion section:

"This is consistent with Jakobs et al. (2021), who found that the snowmelt-albedo feedback begins to strengthen around -12 ∘C and peaks between -9 and -7 ∘C. At higher temperatures, additional albedo-lowering processes become important, including the transition from snowfall to rainfall and increasing exposure of bare ice."